# Hypothermia Therapy for Traumatic Spinal Cord Injury: An Updated Review

**DOI:** 10.3390/jcm11061585

**Published:** 2022-03-13

**Authors:** Seth C. Ransom, Nolan J. Brown, Zachary A. Pennington, Nikita Lakomkin, Anthony L. Mikula, Mohamad Bydon, Benjamin D. Elder

**Affiliations:** 1Department of Neurological Surgery, Mayo Clinic, Rochester, MN 55902, USA; ransom.seth@mayo.edu (S.C.R.); pennington.zachary@mayo.edu (Z.A.P.); lakomkin.nikita@mayo.edu (N.L.); mikula.anthony@mayo.edu (A.L.M.); bydon.mohamad@mayo.edu (M.B.); 2Department of Neurological Surgery, University of California, Irvine, CA 92093, USA; nolanb@hs.uci.edu

**Keywords:** hypothermia, trauma, spine, recovery

## Abstract

Although hypothermia has shown to protect against ischemic and traumatic neuronal death, its potential role in neurologic recovery following traumatic spinal cord injury (TSCI) remains incompletely understood. Herein, we systematically review the safety and efficacy of hypothermia therapy for TSCI. The English medical literature was reviewed using PRISMA guidelines to identify preclinical and clinical studies examining the safety and efficacy of hypothermia following TSCI. Fifty-seven articles met full-text review criteria, of which twenty-eight were included. The main outcomes of interest were neurological recovery and postoperative complications. Among the 24 preclinical studies, both systemic and local hypothermia significantly improved neurologic recovery. In aggregate, the 4 clinical studies enrolled 60 patients for treatment, with 35 receiving systemic hypothermia and 25 local hypothermia. The most frequent complications were respiratory in nature. No patients suffered neurologic deterioration because of hypothermia treatment. Rates of American Spinal Injury Association (AIS) grade conversion after systemic hypothermia (35.5%) were higher when compared to multiple SCI database control studies (26.1%). However, no statistical conclusions could be drawn regarding the efficacy of hypothermia in humans. These limited clinical trials show promise and suggest therapeutic hypothermia to be safe in TSCI patients, though its effect on neurological recovery remains unclear. The preclinical literature supports the efficacy of hypothermia after TSCI. Further clinical trials are warranted to conclusively determine the effects of hypothermia on neurological recovery as well as the ideal means of administration necessary for achieving efficacy in TSCI.

## 1. Introduction

Traumatic spinal cord injury (TSCI) affects 17,800 Americans annually, an incidence that has remained relatively stable over the past three decades [1,2,3]. TSCI is associated with significant increases in care costs, reductions in life expectancy, and reductions in quality of life. Notably, <1% of TSCI victims will be discharged from the hospital with normal neurological function, with approximately 1/3 having complete tetraplegia or paraplegia [1].

Acute SCI pathogenesis is characterized by primary and secondary injury. Primary injury is that which occurs from the initial physical forces of the traumatic event and is often irreversible [4,5]. Secondary injury, in contrast, results from subsequent expansion of the damage [6,7]. It is thought to be, at least in part, preventable, and to stem from multiple mechanisms, including oxidative damage, blood–brain barrier disruption, intracellular hypercalcemia, and neurotransmitter excitotoxicity, which cumulatively drive further tissue destruction [8].

The natural history of SCI suggests that recovery peaks at 3–6 months, though the degree of neurologic improvement is highly varied [9]. Injury severity is commonly graded using the American Spinal Injury Association Impairment Scale (AIS): grade A (no motor or sensory preservation), grade B (sensory function but no motor preservation), grade C/D (some motor function), and grade E (normal motor and sensory function) [10]. A comprehensive database analysis suggests that only 30% of individuals with complete cervical spinal cord injury (AIS grade A) will improve to grade B or better. Among this group, only 30% will recover 2 or more levels below the spinal cord lesion [11]. Due to the significant disability seen in these patients, there is continued interest in identifying therapies to bolster neurologic recovery.

One therapy of recent interest is hypothermia, which regained attention in 2006 following a high-profile case report describing its use in a professional football player who had sustained an AIS grade A injury from a C3/4 fracture dislocation [12]. Over the coming months, the patient made a significant neurological recovery to AIS grade D. This case revived widespread interest in the use of hypothermia for TSCI [13]. These endeavors are supported by a plethora of data from preclinical studies investigating the mechanism by which hypothermia improves TSCI recovery. Both local and systemic methods for hypothermia induction have been studied, and each has its own advantages and disadvantages. Systemic cooling—delivered via surface cooling, endovascular heat exchange catheters, or cold intravenous infusion [14,15,16]—is able to produce rapid, whole-body hypothermia [15,17]. Whereas, local cooling, which makes use of either an epidural heat exchanger or subarachnoid cold solution perfusion [18], theoretically facilitates cooling of local, injured tissue without exposing patients to the risks of systemic hypothermia, such as coagulopathy [19]. The goal of the present study is to systematically review the preclinical and clinical investigations of hypothermia as therapy after TSCI and draw a consensus regarding its effects on individual mortality, neurologic functional improvement/deterioration, and complications.

## 2. Methods

The English language medical literature was reviewed using Preferred Reporting Items for Systematic reviews and Meta-Analyses (PRISMA) guidelines to identify all published preclinical and clinical studies published by 17 August 2021 that have assessed the benefits and safety of hypothermia therapy after TSCI. Articles were identified by querying the PubMed, OVID EMBASE, OVID Medline, Web of Science, and Scopus databases using database-specific queries. The query for PubMed was: (hypothermia OR cooling OR “body temperature” OR “temperature control” OR “body temperature control”) AND (“spine trauma” OR “spinal trauma” OR “spinal cord injury” OR “spine injury” OR “spinal injury” OR “vertebral column injury”). Articles yielded from the search were uploaded into the COVIDENCE (Covidence, Melbourne, Australia) system, and these studies underwent title and abstract screening by two independent reviewers, with points of disagreement being resolved by a third reviewer.

Articles were included if they met the following criteria: (1) Study compared subjects treated with hypothermia (cooling to temperature < 36 °C) as a monotherapy or part of a bundled therapy for patients with spinal cord injury. (2) Study employed a human or animal model of traumatic spinal cord injury at any level. (3) Study evaluated one of the following outcomes: mortality, new-onset neurological deficits, neurological deterioration/improvement, or complications. Articles were excluded on full-text review if the study: (1) did not include primary data (i.e., the article was a commentary, letter to the editor, review article, protocol), (2) described fewer than five patients/subjects, (3) a full-text English translation was not available, or (4) the included patient cohort was assessed more thoroughly in a later study by the same authors. Data were extracted using a custom Covidence template. Full-text review and data extraction were performed by two independent reviewers, with a third reviewer serving as a referee in cases of disagreement. Significant heterogeneity in the protocols implemented prevented quantitative analysis.

## 3. Results

The search query identified 1142 articles, of which 57 met the criteria for full-text review. Of these, 28 articles met the criteria for inclusion in the final qualitative analysis: 24 examined preclinical TSCI models and 4 described human clinical series. Of the 29 excluded studies, the most common reasons were (Figure 1): full text was not available online in English language (*n* = 12) and none of the outlined endpoints were examined (*n* = 9). One clinical study was excluded due to the authors reporting more extensive outcomes on the same patient cohort in a later included study.

### 3.1. Preclinical Data

Of the 24 preclinical studies (Table 1), most (*n* = 21) employed a thoracic spinal cord model [5,20,21,22,23,24,25,26,27,28,29,30,31,32,33,34,35,36,37,38,39,40,41,42]. Twelve studies employed local cooling, eleven used systemic hypothermia, and one evaluated both application modalities. In 12 studies, Sprague-Dawley rats were used as the model of interest. Among the 12 studies using exclusively local hypothermia, 8 saw improvement in motor function. The most common means of inducing local hypothermia was epidural cold perfusion. Ha et al. compared neurological outcomes among Sprague-Dawley rats following T9 spinal cord injury that were treated with either 18 °C saline instilled into the epidural space or underwent contusion alone [26]. Animals in the experimental arm were treated for 48 h with a target epidural temperature of 30 °C. At seven days post-injury, treated animals showed significantly better recovery in neurological function, as assessed on the Gale and inclined plane scores. In contrast, Casas et al. failed to demonstrate a significant difference in neurological recovery following T10 injury in female Sprague-Dawley rats treated with 4 °C epidural saline [23]. Animals in the treatment and control arms showed no difference in Basso–Beatie–Bresnahan (BBB) scores at 6 weeks post-injury. Though the reason for the difference in these two studies employing similar injuries is unknown, differences in the injury severity (25 g·cm in Ha et al. vs. 12.5 g·cm in Casas et al.) or in means of assessment could explain the difference [23,26]. Like Casas et al., Teh et al. used a 12.5 g·cm impact to induce T8 injury in Sprague-Dawley rats treated with sham or local epidural heat exchange, targeting a local cord temperature of 30 °C [38]. However, unlike the aforementioned study, Teh et al. found significant improvement in neurological function on BBB testing at the 6-week follow-up. Importantly, this study employed delayed initiation of therapy, as animals did not start treatment until 2 h post-injury. Across all included studies, none found a significant difference in the rate of neurological deterioration between treatment and control animals.

Of the 11 studies employing systemic hypothermia, 10 employed surface cooling, and the remaining study used a cold air chamber [36]. Interestingly, every preclinical study using surface cooling monotherapy detected improvement in at least one aspect of neurologic function. Seo et al. and Ok et al. tested the effects of systemic hypothermia in a rodent T9 injury model: the former employed cooling for 4 h with surface cooling techniques and the latter 48 h via a cold air chamber [36,37]. Each study found that hypothermia-treated animals had significantly (*p* < 0.05) higher BBB scores at six weeks post-injury. Batchelor et al. studied systemic hypothermia as an adjuvant to surgical decompression in TSCI [22]. Injured animals first underwent systemic hypothermia (target temperature) for 7.5 h, after which surgical decompression was performed, while control animals underwent decompression alone. Animals in the treatment arm had significantly better BBB and ladder-stepping scores as compared to animals that underwent decompression alone.

### 3.2. Clinical Series

Four articles reporting primary data from hypothermia clinical trials were included in this review. An overview of experimental design, inclusion/exclusion criteria, and patient parameters from each study are presented in Table 2. In total, 60 human patients were treated with hypothermia after TSCI, of which 35 received systemic hypothermia and 25 received local hypothermia. Each of the two studies utilizing systemic hypothermia treatment required patients to have an AIS grade A (complete) injury to be eligible for hypothermia therapy [43,44]. Three studies examined hypothermia as an adjuvant to standard care, while the study from Hansebout and Hansebout examined the synergistic effect of local hypothermia and dexamethasone treatment [45]. In all respective cases, intravascular heat exchange catheters were used to induce systemic hypothermia and epidural cooling units were used for local hypothermia. All patients were intubated and sedated based on their clinical condition [43,44,45,46], and those receiving systemic hypothermia were given additional sedation as needed to control shivering [43,44].

Cumulatively, 43% of patients (*n* = 15) with AIS grade A converted to grade B or better following systemic hypothermia therapy and 58% of individuals (*n* = 14) with grade A injury converted following local hypothermia (Figure 2). Of note, the AIS A conversion rates of the 31 patients with cervical injuries who received systemic hypothermia (35.5%) were higher than those from 3 major SCI database control studies of patients with cervical injuries who received standard care (26.1%) [44,47,48,49]. Furthermore, Hansebout and Hansebout reported that 9 out of 14 (64%) patients from their study with cervical TSCI converted from AIS grade A to B or better [45]. None of the patients reported by Gallagher et al. experienced a cervical injury [46]. All clinical studies reported that no neurologic deterioration occurred in any patient as a result of hypothermia treatment.

The most frequent complications were pneumonia (60% following systemic and 28% following local hypothermia) and atelectasis (83% following systemic and 36% following local hypothermia) (Figure 3). No thromboembolic events were seen in the retrospective cohort of 14 patients receiving systemic hypothermia from Levi et al. [44]. However, Dididze et al. reported that 24% (*n* = 5) of patients in their prospective cohort experienced thromboembolic complications: two PEs, one inferior vena cava clot, one DVT of femoral vein, and one clot seen in a subclavian vein [43]. It was noted that the exact same treatment protocols were used for each study, with the exception that Fragmin was used for DVT prophylaxis in the prospective cohort and Lovenox was used in the retrospective cohort [43,44]. The authors also report that one case of ARDS was likely due to aspiration of salt water from drowning and another two were due to aspiration of orogastric fluid during resuscitation [43]. Gallagher et al. decided to terminate their study after 3 out of 5 (60%) patients developed culture-positive wound infections [46]. In this trial, multiple catheters were implanted in the surgical incision site of patients for both local cooling and pathophysiologic monitoring of chemokines/cytokines. On the other hand, only 1 postoperative infection occurred (5%) out of the 20 patients from Hansebout and Hansebout’s study who received local hypothermia [45].

## 4. Discussion

To date, there have been four studies performed which describe clinical outcomes of 60 total patients treated with therapeutic hypothermia. None of the clinical trials reported that a worsening in neurological function occurred because of hypothermia treatment. Overall, these studies found higher AIS conversion rates when adding hypothermia to the TSCI treatment regimen compared to major TSCI database controls. Despite these results, further studies are needed before any definitive conclusions regarding the efficacy of hypothermia in TSCI can be made. Additional investigation into the utility of hypothermia for TSCI has been somewhat tempered by poor outcomes in the literature describing its use for traumatic brain injury (TBI) [50,51]. Like TSCI, TBI was thought to be amenable to hypothermia based on early preclinical and clinical experiences [52,53,54,55]. Nevertheless, despite initial enthusiasm, a recent meta-analysis of the available data found early prophylactic hypothermia to offer no benefit in terms of neurological recovery [51]. Additionally, it was associated with higher rates of complication as compared to standard of care with normothermic temperature goals. For this reason, the most recent version of the Brain Trauma Foundation guidelines recommends against the use of prophylactic hypothermia [56].

### 4.1. Neuroprotective Effects of Hypothermia in Animals

Initial studies performed concerning the neuroprotective role of hypothermia in animals indicated that hypothermia reduces cerebral metabolic oxygen consumption in a canine model [57]. These reports led to the hypothesis that lower temperatures decrease the metabolic demand of the central nervous system, thus protecting it from ischemic damage. Both moderate systemic [25,37,58] and local hypothermia [26,36] treatments exhibit anti-apoptotic and anti-inflammatory effects in preclinical models of TSCI. Some reports indicate that systemic application may suppress extrinsic apoptosis signaling more effectively than localized cooling [36]. Additionally, systemic hypothermia diminishes autophagy and oxidative stress in animal spinal cord contusion models [30,37,59]. Assessment of electrophysiologic, histologic, and functional outcomes following spinal cord injury in rats has demonstrated that those under hypothermic conditions exhibit fewer sequelae compared to those under normothermic conditions following compressive TSCI [60]. Additionally, hypothermia following TSCI in the rat model has been shown to increase normal white matter and gray matter by up to 31% and 38%, respectively [32]. Other observed histologic effects of hypothermia include a four-fold preservation of neurons adjacent to the TSCI epicenter as well as 127% greater sparing of axonal connections supplied by reticulospinal neurons.

### 4.2. New Avenues to Monitor Neurologic Injury

To assess a therapeutic’s efficacy in TSCI remains an extremely difficult task. To date, no statistical conclusions can be drawn on the ability of hypothermia therapy to improve neurologic recovery despite higher AIS conversion rates reported for treated individuals when compared to established control studies from the literature. The broad variation in traumatic injury mechanisms between cases effects neurologic outcomes beyond what can be detected by current imaging standards alone. Indeed, no two TSCI cases are the same. However, novel techniques designed to monitor the inflammatory biomarkers via intrathecal catheter may aid in predicting neurologic recovery after TSCI [61]. Studies from Dalkilic et al. reveal that MRI measures in combination with cerebrospinal fluid (CSF) biomarkers provide the strongest model for classifying baseline AIS grade. However, when predicting AIS conversion, a logistic regression model of CSF biomarkers alone showed 91.2% accuracy and was not aided by the addition of MRI and/or baseline AIS data. Gallagher et al. utilized these techniques in parallel with local hypothermia but stopped the trial due to high infection rates (60%) [46]. Hansebout and Hansebout reported only 1 infection in 20 patients (5%) but initiated treatment sooner than Gallagher and for a shorter duration [45,46]. No studies to date have attempted to monitor CSF biomarkers in patients with TSCI who received systemic hypothermia. However, systemic hypothermia has been unsuccessful in attenuating CSF cytokines after TBI in pediatric patients [62]. Further clinical studies are needed to delineate the relationship between CSF inflammatory markers and neurologic recovery after TSCI.

### 4.3. Safety of Hypothermia Therapy

Current data suggest that hypothermia is not a benign therapy. While most of the complications in the clinical series identified here were attributed to the neurological injury as opposed to hypothermia treatment, patients receiving systemic hypothermia are potentially at higher risk for coagulopathy due to decreased body temperature and presence of an intravenous catheter [63,64]. Notably, a small study (*n* = 10) showed that 50% of patients treated with hypothermia for TBI experienced DVT [65]. However, patients whose catheters were removed within 4 days of insertion had a 33% incidence rate compared to 75% for those removed at later time points. Such risks may potentially be avoidable by implementing a local as opposed to systemic hypothermia regimen. To this end, Levi et al. found similar rates of complications among TSCI patients treated with systemic hypothermia and case-matched controls who did not receive hypothermia therapy [44]. In aggregate, the clinical studies in this review identified venous thromboembolic events in 14% of patients who received systemic hypothermia after TSCI as compared to 24% who were treated with local hypothermia, further supporting the safety of systemic treatment [43]. Nevertheless, local hypothermia confers its own risks, which are in part related to the means in which local cooling is implemented. Gallagher et al. used a cooling catheter placed in the epidural space to effect local hypothermia [46]. Three of their five patients suffered surgical site infection, leading them to prematurely terminate the study. However, in a previous report by the group looking at intradural monitoring for TSCI, no infections were experienced [66]. Consequently, it is unclear if the risk stems from sampling bias or the hypothermia intervention itself.

### 4.4. Future Directions

As demonstrated by the data reviewed here, the limited human studies demonstrate that hypothermia was not associated with any increased risk of new neurological injury or deterioration, though there were increased respiratory risks. Additionally, these data suggest that hypothermia may increase the odds of neurological improvement, by at least one level on the AIS grade, though this data is relatively limited. This by-in-large agrees with the data that are reported for the preclinical, animal literature. Larger, multi-center randomized controlled trials are necessary to draw definitive conclusions on the efficacy of and complication rates associated with hypothermia therapy for TSCI.

## 5. Limitations

There are several limitations to the present study. Preclinical literature has been previously reported to have high rates of non-publication of negative findings [67]. Consequently, the preclinical data presented here may reflect a systemic bias. Indeed, there is a notable degree of heterogeneity present within the studies included in this review. For example, among the animal studies we described, there were significant differences with respect to specific techniques used to model TSCI, animals used, and outcomes measured, among other factors. Additionally, the conclusions drawn from human trials in this review were limited in part by the patient sample size, the retrospective nature of data, and the heterogeneity of patient treatment timelines, as well as the means by which hypothermia therapy was delivered.

## 6. Conclusions

Hypothermia therapies represent a heretofore underexplored therapeutic option for patients with traumatic spinal cord injury. Though the clinical studies at present are extremely limited, they suggest that hypothermia may lead to a small improvement in long-term neurological outcomes, though with some increased respiratory risk. Based on this analysis, hypothermia was found to be a relatively safe therapeutic option with potential to enhance the neurological recovery of individuals who experience a TSCI. Further research is warranted to determine the ideal means of delivering hypothermia as well as the optimum target temperature and treatment duration necessary for achieving efficacy after TSCI in humans.

## Figures and Tables

**Figure 1 jcm-11-01585-f001:**
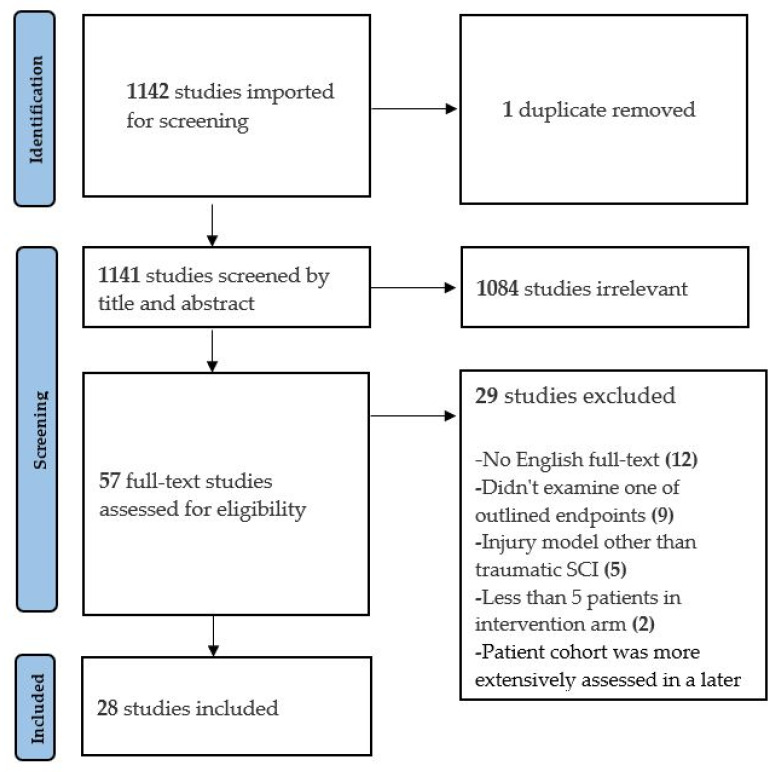
PRISMA flow diagram.

**Figure 2 jcm-11-01585-f002:**
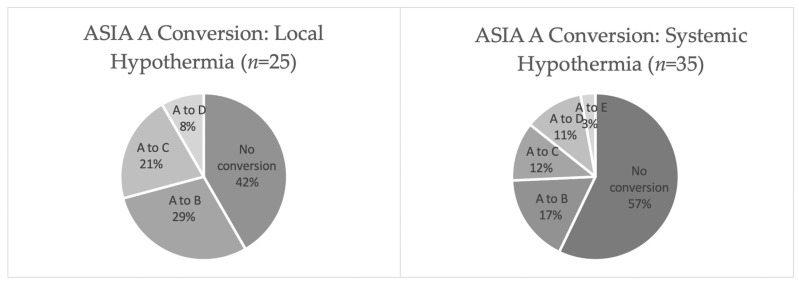
ASIA score conversion in individuals with TSCI who received hypothermia therapy.

**Figure 3 jcm-11-01585-f003:**
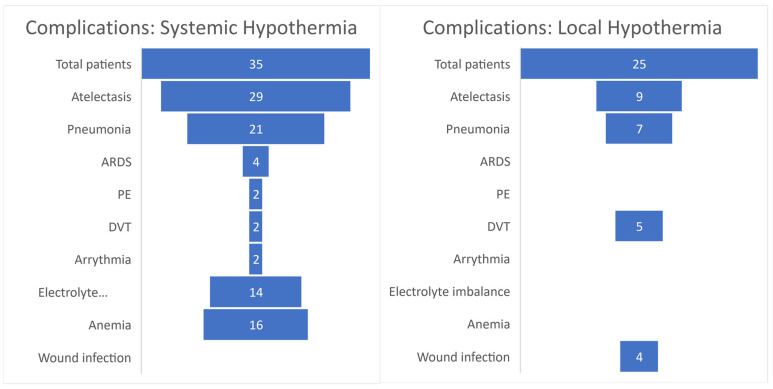
TSCI complication rates in patients treated with hypothermia. ARDS = acute respiratory distress syndrome; PE = pulmonary embolism; DVT = deep vein thrombosis.

**Table 1 jcm-11-01585-t001:** Data overview of 24 animal studies utilizing hypothermia as therapy after TSCI [5,22,23,24,25,26,27,28,29,30,31,32,33,34,35,36,38,39,40,41,42]. NR = not reported; BBB = Basso–Beattie–Bresnahan locomotor rating scale.

Study	Animal	Injury Method	Level of Injury	Hypothermia Method	Treatment Temp/Rate	Target Temp	Treatment Onset	Duration	Other Therapy	Outcomes
Albin et al., 1968	Rhesus monkeys (*n* = 14)	Weight drop (300 g/cf)	T10	Local: subarachnoid heat exchanger	2–5 °C	10 °C	4 h after injury	3 h	None	No deleterious effects were noted. 13 of 14 treated monkeys achieved complete functional recovery.
Barbosa et al., 2014	Wistar rats (*n* = 15 per group)	NYU impactor (25 g/fc)	T9–10	Local: epidural cold solution perfusion	9–10 °C	25 °C	Immediately after injury	20 min	None	No difference in motor outcomes (BBB) were seen between groups.
Batchelor et al., 2010	Female Fischer rats (*n* = 16 per group)	Spinal cord impactor (150 kdyn) and canal spacer	T8	Systemic: surface cooling	NR	33 °C	30 min after injury	7.5 h	Spacer decompression (0, 2, or 8 h after injury)	Hypothermia significantly improved (*p* < 0.005) BBB and ladder-stepping scores in rats decompressed after 8 h.
Casas et al., 2005	Female Sprague-Dawley rats (*n* = 14 per group)	NYU impactor (12.5 g·cm)	T10	Local: epidural saline perfusion	4 °C	20, 30, or 37.5 °C	30 min after injury	3 h	None	No differences in motor outcomes (BBB) were observed between groups.
Dimar et al., 2000	Male Sprague-Dawley rats (*n* = 8 per group)	NYU impactor (25 g·cm), 50% canal spacer, or both	T10	Local: epidural cooling apparatus	19 °C	NR	Immediately after injury	2 h	Spacer decompression (2 h after insertion)	Hypothermia significantly (*p* < 0.05) improved motor (BBB) outcomes in the spacer-only group at 5 weeks post-injury.
Grulova et al., 2013	Male Wistar rats (*n* > 11 per group)	Balloon catheter compression	T8–9	Systemic: surface cooling	NR	31–32 °C	Immediately after injury	30 min	None	Hypothermia improved (*p* < 0.01) urinary scores within 5 days of injury. No difference in motor (BBB) outcomes was observed.
Ha et al., 2008	Male Sprague-Dawley rats (*n* = 8 per group)	NYU impactor (25 g·cm)	T9	Local: epidural saline perfusion	18 °C	30 °C	Immediately after injury	48 h	None	Hypothermia improved (*p* < 0.01) motor function (Gale) and incline plane scores 7 days after injury.
Hosier et al., 2015	Female Long-Evans rats (*n* > 7 per group)	Spinal cord impactor (25 g·cm)	C8; left unilateral	Systemic: surface cooling	−8 °C/h	33 °C	5 min after injury	4 h	None	Hypothermia improved (*p* < 0.05) motor scores 7 days after injury. Beam balance, accelerating rotarod, and BBB scores were greater (*p* < 0.05) in the hypothermia group 6 weeks after injury.
Jorge et al., 2019	Female Sprague-Dawley rats (*n* > 5 per group)	Spinal cord impactor (200 kdyn)	T8	Systemic: surface cooling	NR	32 °C	Beginning of surgery	2 h	None	Hypothermia-treated rats had significantly higher (*p* < 0.01) BBB scores 6 weeks after injury.
Kao et al., 2011	Male Sprague-Dawley rats (*n* = 8 per group)	Aneurysm clip (55 g)	T8–9	Systemic: surface cooling	31 °C	33 °C	Immediately after injury	2 h	None	Hypothermia group had significantly higher (*p* < 0.05) BBB scores 4 days after injury.
Karamouzian et al., 2015	Male Wistar rats (*n* = 20 per group)	Weight drop (25 g·cm)	T8–9	Systemic: surface cooling	NR	33–34 °C	30 min (early) or 3.5 h (late) after injury	3 h	Methylprednisolone (30 mg/kg immediately after injury)	Groups treated with early/late hypothermia, methylprednisolone, or both achieved significantly higher (*p* = 0.05) BBB scores than controls 2–3 weeks after injury. No motor differences were seen among treatment groups.
Kuchner et al., 2000	Female mongrel dogs (*n* > 15 per group)	Epidural balloon inflation (160 mm·Hg)	T13	Local: epidural heat exchanger	NR	3–9 °C	15 min (hypothermia only) or 3.5 h (hypothermia and dexamethasone) after Injury	4 h	Dexamethasone (0.24 mg/kg/day)	Hypothermia (*p* < 0.05), steroid (*p* < 0.05), and dual-therapy (*p* < 0.01) groups had significantly higher motor scores compared to controls at 7 weeks post-injury.
Lo et al., 2009	Female Fischer rats (*n* = 15 per group)	OSU electromagnetic SCI device (3 kdyn)	C5	Systemic: surface cooling	NR	33 °C	After injury	4 h	None	Hypothermia did not improve BBB or incline plane test scores. Hypothermia significantly increased forelimb gripping (*p* < 0.05) and self-supported hanging (*p* < 0.01) 8 weeks after injury
Maybhate et al., 2012	Female Lewis rats (*n* = 10 per group)	NYU impactor (12.5 g·cm)	T8	Systemic: surface cooling	NR	32 °C	2 h after injury	2 h	None	Hypothermia significantly improved (*p* < 0.00004) BBB scores at 4 weeks post-injury.
Morizane et al., 2012	Female Wistar rats (*n* > 8 per group)	NYU impactor (25 g·cm)	T4	Local: extracorpeal thermoelectric cooling device	NR	33 °C	Immediately after injury	48 h	None	Hypothermia-treated rats had significantly better (*p* < 0.01) BBB scores at 8 weeks post-injury
Morochovic et al., 2008	Male Sprague-Dawley rats (*n* = 10 per group)	Epidural balloon catheter	T8–9	Local: surface cooling	−1.4 °C/min	30 °C	25 min after injury	60 min	None	No difference in locomotor performance (BBB) was detected between groups.
Ok et al., 2012	Male Sprague-Dawley rats (*n* = 8 per group)	NYU impactor (25 g·cm)	T9	A. Local: epidural water infusion B. Systemic: cold air chamber	A. 20 °C B. −0.25 °C/min	A. 28 °C B. 32 °C	A. After injury B. After waking from anesthesia	48 h	None	6 weeks after SCI, both local and systemic hypothermia groups had significantly higher (*p* < 0.05) BBB scores.
Seo et al., 2015	Male Sprague-Dawley rats (*n* = 27 per group)	NYU impactor (25 g·cm)	T9	Systemic: surface cooling	NR	30–32 °C	After injury	4 h	None	6 weeks after SCI, hypothermia groups had significantly better (*p* < 0.05) BBB scores than control.
Tator et al., 1973	Rhesus monkeys (*n* = 10 per group)	Inflatable cuff (350–400·mm Hg)	T9	Local: subarachnoid cold solution perfusion	5 °C	NR	3 h after injury and immediately after durotomy	3 h	Durotomy (immediately before hypothermia)	Normothermia-durotomy group recovered significantly better than nonperfused no durotomy group after 400 mm·Hg injury. Hypothermia treatment did not have this effect.
Teh et al., 2018	Sprague-Dawley rats (*n* = 7 per group)	NYU impactor (12.5 g·cm)	T8	Local: epidural heat exchange	−0.5 °C/min	30 °C	2 h after injury	5 or 8 h	None	BBB scores were significantly higher in both 5 h (*p* = 0.001) and 8 h (*p* = 0.006) hypothermia groups compared to control after 6 weeks.
Thienprasit et al., 1975	Adult cats (*n* > 5 per group)	Balloon catheter inflation	L2	Local: epidural saline perfusion	15 °C	NR	6 h after injury	2 h	Decompressive laminectomy (6 h after injury but before cooling)	Animals whose cortical-evoked responses failed to reappear within 6 h of injury had significantly better recovery (*p* < 0.01) after hypothermia and laminectomy compared to laminectomy alone.
Westergren et al., 2000	Male Sprague-Dawley rats (*n* > 5 per group)	Blocking weight technique (50 g)	T7–8	Systemic: surface cooling	NR	30 °C	After injury	2 h	None	Animals treated with hypothermia performed better on inclined plane test at 2 weeks post-injury. No differences in motor function (Gale) scores were detected.
Xu et al., 2016	Male Sprague-Dawley rats (*n* = 7 per group)	Aneurysm clip (10 g)	T10	Local: epidural saline infusion	4 °C	18 °C	Immediately after injury	2 h	None	Hypothermia group achieved significantly higher (*p* < 0.05) BBB scores than control up to 3 weeks after injury.
Yu et al., 2000	Female Sprague-Dawley rats (*n* > 7 per group)	NYU impactor (12.5 g·cm)	T10	Systemic: surface cooling	NR	32 °C	30 min after injury	4 h	None	BBB scores were significantly higher (*p* = 0.0024) 6 weeks after injury in hypothermia group compared to normothermia.

**Table 2 jcm-11-01585-t002:** Overview of four clinical trials using hypothermia as therapy after traumatic spinal cord injury [43,44,45,46]. AIS = ASIA (American Spinal Injury Association) Impairment Scale; GCS = Glasgow Coma Scale.

Study	Design	Total Patients	Inclusion Criteria	Exclusion Criteria	Hypothermia Method	Treatment Temp/Rate	Target Temp	Time to Treatment	Duration	Additional Therapy	Conclusions
Dididze et al., 2013	Case-controlled study	35 (14 from Levi et al., 2010)	1. 18–65 years of age 2. AIS grade A 3. GCS = 15 4. Non-penetrating injury 5. Required urgent surgical reduction	1. Age < 18 or >65 years 2. AIS grade B, C or D 3. Hyperthermia (>37 °C) on admission 4. Severe systemic injury 5. Cord transection 6. Intubation and sedation before initial neurological exam 7. Improved neurological exam within 12 h of injury 8. Other major comorbidities	Systemic: intravascular heat exchange catheter	−2.5 °C/h	33 °C	Average = 7.76 ± 1.09 h	48 h	None	15 out of 35 total patients (43%) improved at least one AIS grade. When excluding patients who converted from AIS A within 24 h, 35.5% (11 out of 31) improved at least one grade. A similar number of respiratory complications occurred in both retrospective (*n* = 14) and prospective (*n* = 21) groups. Overall, 14.2% of patients experienced thromboembolic complications.
Hansebout and Hansebout, 2014	Prospective case series	20	1. Age 16–65 years 2. Alert and cooperative 3. Clinically complete cord injury	1. Motor or sensory function below the level of cord injury 2. Perianal sensation 3. Anal sphincter contraction	Local: epidural cooling unit	3 °C	Dural temp = 6 °C	Average = 7.1 h	Average = 3.7 h	Dexamethasone (6 mg every 6 h for 11 days)	All patients initially had AIS grade A impairment. Of the 20 total patients, 13 (65%) improved at least one AIS score. The most frequent complications were respiratory in nature (45% atelectasis and 35% pneumonia). The overall incidence of thromboembolic events was 25%.
Gallagher et al., 2020	Prospective cohort study	5	1. AIS grade A–C 2. Age 18–70 years 3. Surgery within 72 h of injury 4. Thoracic injuries	1. Other major comorbidities or concurrent injuries 2. Penetrating TSCI 3. No consent	Local: epidural cooling catheter	−0.8 °C/h	Dural temp = 33 °C	Average = 70.4 ± 9.3 h	12 h	None	The study was terminated after 3 out of 5 total patients experienced delayed wound infections. Four patients were initially AIS grade A on admission, of which only 1 improved to AIS B.
Levi et al., 2010	Retrospective comparative case series	14	1. Age 16–65 years 2. AIS grade A 3. Nonpenetrating injury 4. Patients requiring immediate surgical reduction	1. AIS grade B, C, or D 2. Hyperthermia on admission (>38.5 °C) 3. Severe systemic injury 4. Spinal cord transection 5. Improvement in neurologic exam within 12 h of injury 6. Other major comorbidities	Systemic: intravascular heat exchange catheter	−0.5 °C/h	33 °C	Average = 9.17 ± 2.24 h	Average = 47.6 ± 3.1 h	None	6 of the 14 (42.8%) total patients improved their AIS scores. The most frequent complications were respiratory in nature. No thromboembolic complications were reported. A similar number of complications were observed in 14 case-matched control TSCI patients.

## Data Availability

Not applicable.

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
