# Peer review of "Hypothermia Therapy for Traumatic Spinal Cord Injury: An Updated Review"

_jcm, 2022, doi:10.3390/jcm11061585_

Round 1
Reviewer 1 Report
The authors summarize the current literature on the use of hypothermia in TSCI. Predominantly, these are preclinical studies (24). Data from clinical studies on the use of hypothermia in patients is insufficient, which the authors clearly state in their review. The paper is well written, is very well structured and seems logical overall. A clinical recommendation cannot be derived for the above reasons. I noticed some minor aspects:
- The authors describe in section 3.1 that cooling provides an advantage for motor function recovery. It would be interesting for the reader to know exactly how distinct the benefit / neurological rehabilitation potential was.
- How was hypothermia therapy tolerated by the patients? Did they have to be extra sedated / intubated during systematic cooling?
- Small comment on section 4.4: I think rather a RCT is needed for future to objectify the exact benefits on the one hand and the complication rates on the other hand.
Author Response
Reviewer 1,
Thank you for your feedback and critiques. Our responses to your suggestions are as follows:
- It would be interesting for the reader to know exactly how distinct the benefit/neurological rehabilitation potential was.
- While we agree this is a clinically important question, it is not of high priority from a preclinical standpoint due to the variability between SCI models and methods of behavioral assessment not to mention the poor translatability of TSCI therapeutics. Moreover, the robustness of a preclinical SCI therapeutic is almost always mitigated when translated to human trials due the increased severity of SCI pathology in larger animals, i.e. humans. However, this question is of extreme importance in the clinical setting which we believe this paper focuses on in the sections thereafter by trying to glean as much a possible from the limited human data. No adjustments were made to the manuscript to address this comment.
- How was hypothermia therapy tolerated by the patients? Did they have to be extra sedated/intubated during systemic cooling?
- None of the clinical studies commented on patient comfort in response to therapy. However, they did discuss the protocol for intubation/sedation as well as additional sedation for systemic cooling. We added the following to address this comment in lines 192-194: "All patients were intubated and sedated based on their clinical condition, and those receiving systemic hypothermia were given additional sedation as needed to control shivering."
- I think rather a randomized control trial is needed to objectify the exact benefits on the one hand and the complication rates on the other.
- We agree with this. Changes were made in lines 311-312 as follows: "Larger, multi-center randomized controlled trials are necessary to draw definitive conclusions on the efficacy and complication rates of hypothermia therapy for TSCI."
Reviewer 2 Report
Very nice work. Just a single minor comment is would be nice to talk about the biomechanics side as a subsection so the biomedical engineer readers would also benefit from the great literature review provided here.
Author Response
Reviewer 2,
Thank you for your comments and critique. Our response to your suggestion is as follows:
- It would be nice to talk about the biomechanics side as a subsection so the biomedical engineer readers would benefit from the great literature review provided here.
- While we agree it would be interesting to know if hypothermia therapy has any effects on the biomechanics associated with SCI recovery, the authors agree this topic is out of the scope of this review. This paper focuses on the neurologic efficacy and safety of hypothermia therapy in TSCI. Additionally, we did not see any data in the literature that suggested hypothermia therapy has an effect on the biomechanics of spine movement/recovery within the context of SCI. Therefore, we did not add any text in response this comment.